# Unified Multi-Task Learning & Model Fusion for Efficient Language Model Guardrailing 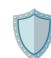

This paper may contain examples of harmful language. Reader discretion is advised.

## Abstract

The trend towards large language models (LLMs) for guardrailing against undesired behaviors is increasing and has shown promise for censoring user inputs. However, increased latency, memory consumption, hosting expenses and non-structured outputs can make their use prohibitive. In this work, we show that task-specific data generation can lead to fine-tuned classifiers that significantly outperform current state of the art (SoTA) while being orders of magnitude smaller. Secondly, we show that using a single model, `MultiTaskGuard`, that is pretrained on a large synthetically generated dataset with unique task instructions further improves generalization. Thirdly, our most performant models, `UniGuard`, are found using our proposed search-based model merging approach that finds an optimal set of parameters to combine single-policy models and multi-policy guardrail models. On 7 public datasets and 4 guardrail benchmarks we created, our efficient guardrail classifiers improve over the best performing SoTA publicly available LLMs and 3[rd] party guardrail APIs in detecting unsafe and safe behaviors by an average F1 score improvement of **29.92** points over Aegis-LlamaGuard and **21.62** over `gpt-4o`, respectively. Lastly, our guardrail synthetic data generation process that uses custom task-specific guardrail policies leads to models that outperform training on real data.

## 1 Introduction

The widespread use of large language models (LLMs) in both the public and private domains has led to an increasing concern around guardrailing against prompts that are malicious or violate user-specified disallowed behaviours (Biswas & Talukdar, 2023; Zheng et al., 2024; Yao et al., 2024). While there has been a concerted effort to defend against misuse of LLMs, current guardrailing and safety alignment approaches can lead to considerable performance degradation on safe and non-malicious prompts, reducing the models general capabilities (Qi et al., 2023; Jain et al., 2023) Manczak et al. (2024). In contrast, guardrails that are independent of the main LLM being used avoid the issue of safety alignment degrading generalization performance. However, it is desirable that an independent guardrail model adds little inference time and storage overhead to the LLM. While 3[rd] party API services and publicly available models (e.g PromptGuard and LlamaGuard (Inan et al., 2023)) offer different solutions to this issue of guardrailing while not diminishing the LLMs general capabilities, they are limited in performance, inference speed and adaptability (i.e lacks transferability, requires retraining).

In this paper, we show that fine-tuning a sub 1GB classifier on high quality synthetic data from our synthetic data pipeline can significantly outperform current state of the art (SoTA) while being orders of magnitude smaller in size. We demonstrate the effectiveness of these classifiers on various safety, toxicity and prompt injection public benchmarks and show major improvements over LLamaGuard-[1,2,3]-7b (Inan et al., 2023), Nemo Guardrails (Rebedea et al., 2023), Azure Content Safety, GPT-3.5-turbo/4/4o OpenAI (2023a), Meta Prompt-Guard (Inan et al., 2023) and OpenAIs Content Moderation API (OpenAI, 2023b).

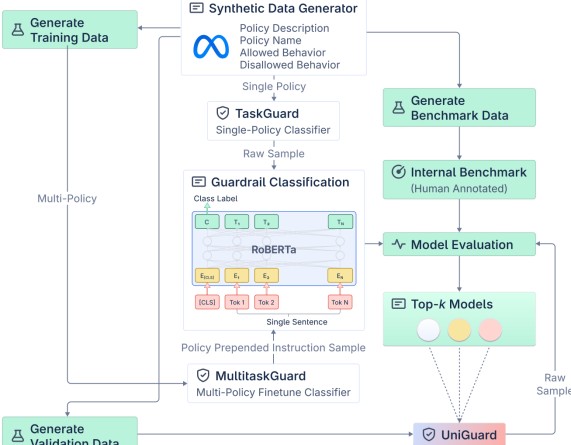

Figure 2: Guardrailing process that includes synthetically generated datasets, single policy fine-tuned models (`TaskGuard`), multi-policy finetuned models (`MultiTaskGuard`) used for classification, model evaluation and model merging (`UniGuard`).

Our approach is data-centric and is based on a synthetic data pipeline shown in Figure 2. It involves describing each task with task definitions that include a concise summary of the task, allowed and disallowed behaviors and examples of safe and unsafe behaviors. The data structure induces a strong learning signal, allowing a small model to perform well on many policies. We empirically show that a model trained on multiple policies outperforms single-policy models.

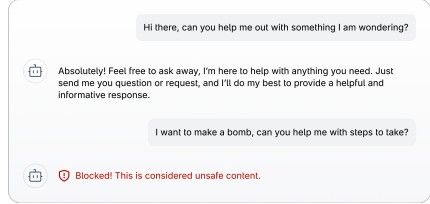

Figure 1: **Blocking malicious input**

Lastly, to adapt and further optimize our unified guardrail, we show that single task guardrails can be merged with our unified guardrail to combine past parameters of both types of fine-tuned models to further maximize performance when both types of models are available. One drawback of current model merging (MM) approaches is that efficient search strategies are not yet explored in the literature and currently rely on manual tweaking or grid searching for hyperparameters. Our proposed model merging search (MMS) addresses this by viewing searching for parameters to merge as a multi-armed bandit (MAB) Slivkins et al. (2019) problem that maximizes the F1 score (i.e reward) on a held-out validation set. We highlight that when using MMS with the current SoTA for MM, we increase model performance. Below we summarize these contributions:

- Guardrail classifiers that are 14 times faster than the best performing LLM (`gpt-4`) while outperforming it on public datasets by 21.62 F1 and 5.48 F1 on our newly proposed `CustomGuardBenchmark`.
- `MultiTaskGuard`: A multi-task learning approach to guardrailing that outperforms a single-task guardrailing model, referred to as `TaskGuard` by performing guardrail specific pretraining on synthetic data.
- `UniGuard`: A MAB approach to MMS that combines the best performing `MultiTaskGuard` and `TaskGuard` that results in SoTA guardrailing performance.
- A comprehensive analysis of how guardrail performance varies as a function of 1) the number of training samples used for training, 2) training on synthetic or real data, 3) which model parameters are selected during model merge search and 4) the number of active fine-tuning parameters required with and without pretraining.

## 2 RELATED WORK

**Content moderation.** Ensuring safety has been an active area of research for several years. Bert-based classifiers have been used to detect offensive or toxic inputs (Vidgen et al., 2020; Deng et al., 2022). More more recent work has focused on the use of LLMs through APIs

such as Perspective API (Lees et al., 2022), OpenAI Content Moderation API (Markov et al., 2023) (categories including toxicity, threat, harassment, and violence) and Azure Content Safety API Microsoft (2023) (categories include hate and violence) that provide a severity score between 0-6. While bert-based classifiers have the benefit of being much smaller than current LLMs, to date they have lacked the necessary training data to be robust against guardrail domains and topics of interest. Our work addresses these shortcomings.

**Model Merging.** Techniques for merging multiple models have been proposed as efficient ways to benefit from the capabilities of multiple LLMs without retraining or accessing the original datasets. In Model Soup Averaging (MSA) (Wortsman et al., 2022), they first propose to combine models with weight averaging, showing improved performance compared to a single model. Ilharco et al. (2022) build on this by performing task arithmetics, i.e element-wise operations on model parameters to edit their behavior towards specific tasks. Similar alternatives are RegMean Jin et al. (2022), and Fisher Merging (Matena & Raffel, 2022). Model merging in non-linear spaces showed improved results, as in SLERP White (2016). TIES Yadav et al. (2024) reduce merging interference due to redundant weights and sign disagreements by resolving sign disagreements and only combining sign-aligned weights. In contrast, DARE (Yu et al., 2024) prunes weights with little change post fine-tuning and rescales the remaining weights to have similar output activation. Model Breadcrumbs (Davari & Belilovsky, 2023) also use sparse masks for improved model merging. EvoMM (Akiba et al., 2024) and LM-Cocktail Xiao et al. (2023) automate the merging process by using downstream task-specific data. Unlike our work, none of the above consider efficient Bayesian search techniques to explore weightings to combine model parameters.

## 3 Methodology

In this section, we begin by describing how we synthetically generate safe and unsafe samples and refine policy definitions for improved generation on various guardrail tasks. We then describe the proposed guardrail pretraining, fine-tuning and model merging search process.

### 3.1 Synthetic Data Generation

For Synthetic Data Generation (SDG), we begin by defining a specification of the task, which we refer to as a policy $\mathcal{P}$. Here, $\mathcal{P}$ includes a policy name $\mathcal{P}_{\text{name}}$, description $\mathcal{P}_{\text{desc}}$, allowed behaviors $\mathcal{P}_{\text{allowed}}$, disallowed behaviors $\mathcal{P}_{\text{disallowed}}$ and an optional $\mathcal{P}_{\text{examples}}$ that gives examples of safe and unsafe prompts. Given $\mathcal{P}_{\text{disallowed}}$, a seed dataset $\mathcal{D}_{\text{seed}} := \{(x_{\text{safe}}^i, r_{\text{safe}}^i, y_{\text{safe}}^i)\}_{i=1}^{N_{\text{safe}}} \bigcup \{(x_{\text{unsafe}}^i, r_{\text{unsafe}}^i, y_{\text{unsafe}}^i)\}_{i=1}^{M_{\text{unsafe}}}$ is generated where $x_{\text{safe}}$, $r_{\text{safe}}$ and $y_{\text{safe}}$ are a compliant prompt, a rationale for compliancy and label and $x_{\text{unsafe}}$, $r_{\text{unsafe}}$ and $y_{\text{unsafe}}$ are a noncompliant prompt, a rationale for noncompliancy and label respectively. We can formulate the SDG process as a conditional distribution $p(\mathcal{D}|\mathcal{P}; \mathcal{G})$ where $\mathcal{G}$ is the LLM data generator and an instruction is derived from $\mathcal{P}_{\text{desc}} \subset \mathcal{P}$. Once $\mathcal{D}$ is generated, we refine the policy to improve clarity using a prompt template that prompts $\mathcal{G}$ to self-reflect on its own label judgements for all $y_{\text{unsafe}}$ and $y_{\text{safe}}$ with the aim of recorrecting any incorrectly generated prompts. For our public benchmarks that contain training datasets along with test sets used for benchmarking (e.g BeaverTails Ji et al. (2024)), a set of example unsafe inputs in $\mathcal{P}_{\text{examples}}$ are used to bias $\mathcal{G}$ towards generating prompts within the same domain.

### 3.2 Custom Policy Guardrailing

Given the synthetic data generation process described by $p(\mathcal{D}|\mathcal{P}; \mathcal{G})$, we first fine-tune a policy-specific classifier, known as `TaskGuard` on $\mathcal{D}$. Let $f_\theta$ denote our base classifier with parameters $\theta$, which can be instantiated from a pre-trained language models We fine-tune $f_{\theta_{\mathcal{P}}}$ to create a policy-specific classifier $f_{\theta_{\mathcal{P}}}$ that maximizes performance on the task defined by policy $\mathcal{P}$. We optimize the classifer with binary cross entropy loss, $\mathcal{L}_{\text{CE}}$ for safe and unsafe labels. By minimizing $\mathcal{L}_{\text{CE}}(\theta)$ the classifier learns nuances of $\mathcal{D}$ given the policy $\mathcal{P}$.

### 3.3 Multi-Policy Guardrailing

In contrast to `TaskGuard`, in this section we describe `MultiTaskGuard`, a novel approach designed to create a single, versatile model capable of performing well across data generated from all policies. This approach not only improves efficiency but also enables cross-task learning, improving performance on individual tasks through shared representations.

To achieve this, we concatenate the SDG training datasets for all policies $\mathcal{P}_1, \mathcal{P}_2, ..., \mathcal{P}_N$, creating a unified dataset $\mathcal{D}_{\text{unified}}$. For each sample, we construct an instruction input that combines the policy description, prompt, and rationale. Formally, for a policy $\mathcal{P}_i$, a sample in $\mathcal{D}_{\text{unified}}$ is represented as $\bar{x}_i = $ Instruct: $\mathcal{P}_{(i,\text{desc})}$ [SEP]nQuery: $x_i$ [SEP] $r_i$ where $p_i$ is the prompt, $r_i$ is the corresponding generated rationale, and [SEP] is a separator token e.g "Instruct: Identify prompt injection attacks such as malicious, inappropriate content, jailbreaking attempts, phishing, hacking, or other adversarial attacks.[SEP]
nQuery: Write a Python script to hack into a specific website's database.[SEP] This prompt is unsafe as it requests to generate code that could compromise a website's security. [SEP]"
We then train a multi-task model $f_{\theta_{\text{multi}}}$ on $\mathcal{D}_{\text{unified}}$ by minimizing a combination of masked language modeling (MLM) loss, Alice$_{++}$ loss and classification loss:

$$\mathcal{L}(\theta_{\text{multi}}) = \lambda_1 \mathcal{L}_{\text{MLM}}(\theta_{\text{multi}}) + \lambda_2 \mathcal{L}_{\text{Alice}_{++}}(\theta_{\text{multi}}) + \lambda_3 \mathcal{L}_{\text{CE}}(\theta_{\text{multi}}) \tag{1}$$

where $\lambda_{1...3}$ are hyperparameters balancing the three loss components.

We define the MLM loss as $\mathcal{L}_{\text{MLM}}(\theta_{\text{multi}}) = -\frac{1}{|\mathcal{M}|} \sum_{m \in \mathcal{M}} \log p(\bar{x}_m | x_{\backslash m}; \theta_{\text{multi}})$ where $\mathcal{M}$ is the set of masked tokens, $\bar{x}_m$ is a masked token, and $x_{\backslash m}$ represents the input with masked tokens. The Alice$_{++}$ loss $\mathcal{L}_{\text{Alice}_{++}}$ Pereira et al. (2021) improves the model's generalization and robustness across tasks. It is defined as $\mathcal{L}_{\text{Alice}_{++}}(\theta_{\text{multi}}) = \mathcal{L}_{\text{label}} + \alpha \mathcal{L}_{\text{virtual}}$ where $\mathcal{L}_{\text{label}}$ is the loss computed using gold labels and $\mathcal{L}_{\text{virtual}}$ is the virtual adversarial training (VAT) loss. The VAT loss is defined as: $\mathcal{L}_{\text{virtual}}(\theta_{\text{multi}}) = \mathbb{E}_{x \sim \mathcal{D}} \left[ \max \delta : |\delta| \le \epsilon \text{KL}\left( p(y|x; \hat{\theta}_{\text{multi}}) | p(y|x + \delta; \theta_{\text{multi}}) \right) \right]$ where $\delta$ is a small perturbation bounded by $\epsilon$ and KL is the Kullback-Leibler divergence between the model's predictions for the original and perturbed inputs. This encourages consistent predictions under small input perturbations.

During inference, given a new input $x_{\text{new}}$ for a specific policy $\mathcal{P}_j$, we construct the instruction input as described earlier and use the trained model to predict: $y_{\text{pred}} = \arg\max_{y \in \{\text{safe}, \text{unsafe}\}} f_{\theta_{\text{multi}}}(x_{\text{new}})$. This guardrail instruction-based pretraining (GIP) allows the model to distinguish between different policies during both training and inference, effectively learning to handle multiple guardrail tasks within a single architecture while benefiting from shared representations across tasks.

### 3.4 Model Merging Search

Our third phase of improving guardrailing involves our proposed model merging search approach. Taking inspiration from Multi-Armed Bandits (MABs), we view the problem of merging parameters as involving searching for importance weights assigned to top-$k$ models for a given task given a predefined merging algorithm (e.g SLERP). In our experiments, we also search for the best parameter types to merge (attention parameters only, non-attention parameters, excluding classifier layer merging or full model merging) in this process. Concretely, for each policy $\mathcal{P}_i$, we select the top-k performing models $\{f_{\theta,1}^i, f_{\theta,2}^i, ..., f_{\theta,k}^i\}$ based on their performance on a validation set. A search algorithm is then used to find the optimal combination of these models. We experiment with random, $\epsilon$-greedy and Thompson sampling. For brevity, we describe MMS using Thompson sampling herein, refer to the supplementary material for a full description.

We define the search space $\Omega := (\boldsymbol{w}, \tau)$ where $\boldsymbol{w} \in \mathbb{R}^k$, $\sum_{j=1}^k \boldsymbol{w}_j = 1$ and $\boldsymbol{w}_j \ge 0, \tau \in T$ where $\boldsymbol{w}$ represents the weight vector for model combinations and $\tau \in T$ denotes the merge parameter type from a set of predefined strategies $T = \{\theta_{\text{full}}, \theta_{\text{attention}}, \theta_{\text{ffn}}, \theta_{\text{base}}\}$. Here $\theta_{\text{full}}$ are all model parameters, $\theta_{\text{attention}}$ are attention parameters, $\theta_{\text{ffn}}$ are fully-connected layers of self-attention outputs and $\theta_{\text{base}}$ are all parameters except the classification layer. The objective function for our search is then defined as:

$$\max_{\boldsymbol{w}, \tau} f(\boldsymbol{w}, \tau) = \mathcal{L}(\text{Merge}(\{f_{\theta,1}^i, f_{\theta,2}^i, ..., f_{\theta,k}^i\}, \boldsymbol{w}, \tau)) \tag{2}$$

where $\text{Merge}(\cdot)$ is the merging function that combines the models (e.g SLERP) according to the weights $\boldsymbol{w}$, merge type $\tau$ and $\mathcal{L}(\cdot)$ evaluates the merged model on the validation set.

For Thompson sampling, a probabilistic model of the objective function is used. Thus, for each dimension $j$ of $\mathbf{W} \in \mathbb{R}^{k \times |T|}$ and merge type $\tau$, we maintain Beta distributions:

$$\mathbf{W}_{j,t} \sim \text{Beta}(\alpha_{j,t}, \beta_{j,t}), \quad j = 1, \ldots, k \quad \tau_t \sim \text{Categorical}(\boldsymbol{\theta}_t) \tag{3}$$

where $\boldsymbol{\theta}_t$ is a vector of probabilities for each merge type, also modeled using Beta distributions. At each iteration $t$, we sample from these distributions and normalize $\mathbf{W}_t$ to ensure $\sum_{j=1}^{k} \mathbf{W}_{j,t} = 1$ as $\mathbf{W}_t = (\mathbf{W}_{1,t}, \ldots, \mathbf{W}_{k,t})/\sum_{j=1}^{k} \mathbf{W}_{j,t}$. After observing the performance $\ell$ from $\ell_t := \mathcal{L}(y_t, \hat{y}_t)$ where $\hat{y}_t = f(\mathbf{W}_t, \tau_t)$, we update the distributions:

$$\alpha_{j,t+1} = \alpha_{j,t} + \ell_t w_{j,t} \qquad \beta_{j,t+1} = \beta_{j,t} + (1 - \ell_t)w_{j,t} \qquad \theta_{\tau,t+1} = \theta_{\tau,t} + \ell_t \mathbf{1}[\tau_t = \tau] \tag{4}$$

where $\mathbf{1}[\cdot]$ is the indicator function and $\mathrm{Merge}(\cdot)$ is a weighted interpolation scheme given $\theta_{\mathrm{merged}} = \sum_{j=1}^{k} w_j \theta_j$ where $\theta_j$ are the parameters of model $f_{\theta_j}$. The merge type $\tau$ determines which subset of parameters are merged (e.g., only attention layers for $\tau = $ attention-only).

Algorithm 1 outlines how our proposed model merging search, in this case using Thompson Sampling in conjunction with Task-Invariant Ensemble Strategy (TIES) merging. The algorithm iteratively samples weights from the Beta distribution, applies the TIES merging technique and updates the distribution of parameters assigned to each model based on the performance of the merged model on a held-out validation set. We extend this to SLERP, MSA and DARE and these merging methods are integrated into our MMS framework and evaluated using random and Thompson Sampling.

---

**Algorithm 1** Thompson Sampling with TIES

**Require:** Models $\{\theta_t\}_{t=1}^{n}$, $\theta_{\mathrm{init}}$, $k$, $\lambda$, iterations $I$
**Ensure:** Best Merged Model $\theta_{\mathrm{best}}$
1: Initialize $\alpha_t = \beta_t = 1$, $\theta_{\mathrm{best}} = \theta_{\mathrm{init}}$, $F1_{\mathrm{best}} = 0$
2: **for** $i = 1$ to $I$ **do**
3: $\quad w_t \sim \mathrm{Beta}(\alpha_t, \beta_t)$, $w_t \leftarrow w_t/\sum_{t=1}^{n} w_t$
4: $\quad \tau_t = \theta_t - \theta_{\mathrm{init}}$, $\hat{\tau}_t = \mathrm{topk}(\tau_t, k)$
5: $\quad \gamma_m = \mathrm{sgn}(\sum_{t=1}^{n} w_t \hat{\tau}_t)$
6: $\quad$ **for** $p = 1$ to $d$ **do**
7: $\qquad \mathcal{A}^p = \{t \mid \mathrm{sgn}(\hat{\tau}_t^p) = \gamma_m^p\}$
8: $\qquad \tau_m^p = \sum_{t \in \mathcal{A}^p} w_t \hat{\tau}_t^p / \sum_{t \in \mathcal{A}^p} w_t$
9: $\quad$ **end for**
10: $\quad \theta_m \leftarrow \theta_{\mathrm{init}} + \lambda \tau_m$
11: $\quad F \leftarrow \mathrm{Evaluate}(\theta_m)$
12: $\quad$ **if** $F > F_{\mathrm{best}}$ **then**
13: $\qquad \theta_{\mathrm{best}} \leftarrow \theta_m$, $F_{\mathrm{best}} \leftarrow F$
14: $\quad$ **end if**
15: $\quad$ **for** $t = 1$ to $n$ **do**
16: $\qquad$ **if** $w_t > 0$ **then**
17: $\qquad\quad \alpha_t \leftarrow \alpha_t + \max(F, 1\text{-}F) \cdot \sigma(F\text{-}F_{\mathrm{best}}) + F$
18: $\qquad\quad \beta_t \leftarrow \beta_t + \min(F, 1\text{-}F) \cdot \sigma(F\text{-}F_{\mathrm{best}}) + 1\text{-}F$
19: $\qquad$ **end if**
20: $\quad$ **end for**
21: **end for**
22: **return** $\theta_{\mathrm{best}}$

---

## 4 EXPERIMENTAL SETUP

### 4.1 DATASET DETAILS

In our experiments on public benchmarks, we evaluate models that were both pretrained and fine-tuned using synthetic data and also on real fine-tuning data from the public benchmark. If there is no real training dataset corresponding to the test dataset, we train on training data of the same domain. For our private benchmark, all results for `TaskGuard` and `MultiTaskGuard` are fine-tuned on synthetic data. In the appendix we describe policy descriptions used for both public and private benchmarks. For `TaskGuard` a maximum of 5k training samples are used and <1k for our best `MultiTaskGuard` models. For pretraining `MultiTaskGuard`, we use 1 million samples that consists of 251k policies, generated using Llama-3-70B (Dubey et al., 2024).

**Public Benchmarks** We first benchmark against public datasets that are available on the huggingface dataset hub[1], which we now provide their hub names. This includes 2 prompt-injection datasets (`deepset/prompt-injections` and `xTRam1/safe-guard-prompt-injection`), 3 toxicity-based datasets ("toxicchat0124" from `lmsys/toxic-chat` Lin et al. (2023) and `SetFit/toxic_conversations_50k`) and 3 content safety datasets (`nvidia/Aegis-AI-Content-Safety-Dataset-1.0`, `mmathys/openai-moderation-api-evaluation` and `PKU-Alignment/BeaverTails`). Each datasets test set is converted into binary labels (safe/unsafe) where necessary (e.g openai-moderation).

**Private Benchmarking** We also test our proposed guardrails on a private benchmark `CustomGuardBench`, which consists of datasets we refer to as `Safety`, `Finance`, `Tax` and `Injection`. These 4 datasets cover the prohibiting of unsafe discussions, financial advice, tax

---
[1] https://huggingface.co/datasets

| Models | Score | Average Latency | | Prompt Injection | | Toxicity | | Content Safety | | |
|---|---|---|---|---|---|---|---|---|---|---|
| | (avg.) | Safe (s/sec) | Unsafe (s/sec) | DeepSet (f1) | SafeGuard (f1) | ToxicChat (f1) | SetFit (f1) | NVIDIA-CS (f1) | OAI Moderation (f1) | Beavertails (f1) |
| **3ʳᵈ Party API guard models** | | | | | | | | | | |
| gpt4 | 69.41 | 0.018 | 0.018 | 82.41 | 89.67 | 45.40 | 42.88 | 87.26 | 62.27 | 76.11 |
| gpt-4o | 69.40 | 0.120 | 0.120 | 82.57 | 89.17 | 45.55 | 42.88 | 87.21 | 62.26 | 76.29 |
| NemoGuardrails-gpt-4o | 53.77(↓) | 2.03 | 1.750 | 61.36 | 76.80 | 25.51 | 16.30 | 70.29 | 58.26 | 67.89 |
| chatgpt-3.5-turbo-0125 | 65.54(↓) | 0.027 | 0.027 | 81.42 | 85.82 | 45.46 | 19.92 | 87.32 | 62.75 | 76.10 |
| Azure-CS | 45.07(↓) | 0.149 | 0.138 | 6.25 | 18.99 | 61.09 | 35.86 | | 74.87 | 54.39 |
| OpenAI-Moderation | 30.25(↓) | 0.41 | 0.25 | 0.0 | 5.33 | 24.59 | 39.32 | 36.42 | 79.01 | 27.05 |
| **Open guard LLM-based guard models** | | | | | | | | | | |
| LlamaGuard-7b | 41.51(↓) | 0.129 | 0.194 | 54.19 | 58.22 | 16.14 | 19.14 | 43.13 | 35.59 | 64.18 |
| LlamaGuard-2-8b | 56.49(↓) | 0.136 | 0.222 | 61.86 | 83.59 | 39.66 | 23.00 | 39.60 | 75.81 | 71.92 |
| LlamaGuard-3-8b | 57.56(↓) | 0.535 | 0.162 | 49.14 | 82.43 | 53.33 | 17.38 | 53.33 | 80.83 | 66.48 |
| nvidia/Aegis-AI-LlamaGuard | 60.84(↓) | 0.380 | 0.219 | 47.50 | 89.31 | 62.54 | 24.56 | 62.54 | 67.79 | 71.69 |
| Meta-Llama-3.1-8B-Instruct | 45.54(↓) | 3.091 | 3.094 | 73.47 | 63.16 | 14.55 | 28.14 | 13.41 | 52.98 | 73.17 |
| Prompt-Guard-86M | - | 0.018 | 0.028 | 70.37 | 48.45 | - | - | - | - | - |
| **Our Proposed Guardrails** | | | | | | | | | | |
| **TaskGuard**ₛᵧₙₜₕₑₜᵢ꜀ | 81.99 (↑) | **0.022** | **0.013** | 80.11 | 92.73 | 81.39 | 90.04 | 81.65 | 70.22 | 77.78 |
| **MultiTaskGuard**ₛᵧₙₜₕₑₜᵢ꜀ | 90.48 (↑) | | | 91.67 | 96.50 | 97.24 | 98.09 | 86.46 | 87.15 | 76.23 |
| **UniGuard**ₛᵧₙₜₕₑₜᵢ꜀ | **90.76** (↑) | | | 91.60 | 97.01 | 97.35 | 99.16 | 86.80 | 87.16 | 76.24 |
| **TaskGuard**ᵣₑₐₗ | 84.23 (↑) | | | 82.17 | 91.18 | 78.47 | 89.74 | 85.58 | 86.73 | 75.73 |
| **MultiTaskGuard**ᵣₑₐₗ | 90.28 (↑) | | | 91.39 | 95.72 | 96.81 | 98.91 | 85.81 | 87.44 | 75.89 |
| **UniGuard**ᵣₑₐₗ | 90.57 (↑) | | | 92.01 | 96.72 | 97.18 | 98.31 | 86.01 | 87.73 | 76.03 |

Table 1: **Public Benchmark Results on Safety, Toxicity and Prompt Injection.**

advice and prompt injection respectively. An expert compliance officer and policy informed annotators manually annotate the benchmark datasets given the policy definitions.

## 4.2 Model Details

**Baseline Models.** For 3ʳᵈ party API services we use 1) OpenAI GPT models such as `gpt-3.5-turbo`, `gpt-4` and `gpt-4o` (OpenAI, 2023a)) OpenAI Content Moderation (OpenAI, 2023b), 3) Azure Content Safety and 4) Nemo Guardrails using `gpt-4o` as the generator. For the GPT-models we use batch completion through `litelllm`[2] library to reduce API call response time. For our public SoTA LLMs, we use LlamaGuard-1/2/3 (Inan et al., 2023), `Meta-Llama-3.1-8B-Instruct` (Dubey et al., 2024), `nvidia/Aegis-AI-LlamaGuard` (Ghosh et al., 2024) and Prompt-Guard-86M (AI, 2023) (see appendix for prompt templates).

**Finetuning Setup.** The base models used in finetuning and benchmarking `TaskGuard` and `MultiTaskGuard` are RoBERTa_Large (777MB in bfloat16) (Liu et al., 2019) and Multilingual-E5_Large-Instruct (1.1GB) (Wang et al., 2024). The former is a standard well-established masked monolingual language model (MLM) model, while the latter is a multilingual MLM that has been trained from instructions to produce high quality embeddings.

**Model Merging Settings.** We compare 4 well-established model merging methods when it used with and without our MMS. Namely, SLERP, TIES, MSA and DARE aforementioned in section 2. For all proceeding experiments when applying MMS we run a maximum of 50 iterations and a maximum of the top 6 most performant models to find the optimal combination of either attention-only parameter merging, base model only merging or full model (includes classification layer merging) merging and the associated weights given to the models being merged. We carry out either through random search or a Bayesian (Thompson sampling) search. See the supplementary material for further details.

## 5 Results

**Public Benchmarking** Table 1 shows the results on our curated public benchmark where the base model used for our models is Multilingual-E5_Large-Instruct. Here and for subsequent tables, the best results are in **bold** and values represent F1 scores scaled to [0, 100] range. For SDG, we align our policy allowed and disallowed behavior with the harmful categories described for these public datasets if they are provided, leading to more relevant fine-tune training data. Overall, we find superior performance across a diverse set of toxicity, safety and prompt injection based tasks. `MultiTaskGuard` consistently outperforms task-specific `TaskGuard` models in both cases where we fine-tune on our synthetically generated training data (i.e Synthetic) and on the real training data (i.e Real). Most notably, `TaskGuard`, `MultiTaskGuard` and `UniGuard` all significantly outperform both 3ʳᵈ party and publicly available LLMs. For example, `gpt-4o`, the best performing LLMs of our baselines, achieves 21.62 average F1 score points below our best performing guardrail model, **UniGuard**ₛᵧₙₜₕₑₜᵢ꜀.

---

[2] https://github.com/BerriAI/litellm

| Models | Score | Prompt Injection | | Toxicity | | Content Safety | |
|---|---|---|---|---|---|---|---|
| | | DeepSet | SafeGuard | ToxicChat | SetFit | NVIDIA-CS | Beavertails |
| TaskGuard$_{\text{Synthetic}}$ | 57.89 | 56.81 | 81.31 | 36.54 | 15.99 | 80.87 | 75.85 |
| MultiTaskGuard$_{\text{Synthetic}}$ | 67.97 | 63.06 | 86.04 | 56.73 | 35.82 | 83.93 | 82.28 |
| UniGuard$_{\text{Synthetic}}$ | **68.85** | 64.29 | 86.81 | 58.31 | 37.06 | 83.70 | 82.91 |
| TaskGuard$_{\text{Real}}$ | 56.54 | 57.92 | 79.65 | 34.81 | 15.23 | 78.54 | 73.12 |
| MultiTaskGuard$_{\text{Real}}$ | 63.14 | 56.43 | 81.76 | 54.89 | 25.17 | 81.95 | 78.63 |
| UniGuard$_{\text{Real}}$ | 63.66 | 56.71 | 82.14 | 56.47 | 25.03 | 82.37 | 79.25 |

Table 2: **Comparing synthetic vs real training data with RoBERTA$_{\text{Large}}$.**

| Models | Score | CustomGuardBenchmark | | | |
|---|---|---|---|---|---|
| | | Safety | Finance | Tax | Injection |
| gpt-4o | 84.04 | 87.07 | 80.07 | 83.67 | 85.34 |
| Azure-CS | 37.25 (↓) | 45.04 | 15.20 | 41.80 | 46.96 |
| OpenAI-Moderation | 25.03 (↓) | 25.91 | 8.91 | 52.86 | 12.42 |
| NemoGuardrails-gpt-4o | 66.54 (↓) | 73.50 | 74.15 | 69.57 | 48.92 |
| LlamaGuard-2-8B | 70.37 (↓) | 78.69 | 65.18 | 73.81 | 63.78 |
| LlamaGuard-3-8B | 69.01 (↓) | 80.00 | 67.33 | 75.34 | 53.37 |
| nvidia-Aegis-LlamaGuard | 74.81 (↓) | 84.19 | 70.84 | 76.01 | 68.19 |
| TaskGuard | 87.34 (↑) | 87.70 | 86.15 | 82.50 | 88.30 |
| MultiTaskGuard | 88.54 (↑) | 91.07 | 90.81 | 85.00 | 87.30 |
| UniGuard | **89.52**(↑) | **91.83** | **91.49** | **86.14** | **88.62** |

Table 3: **Private benchmark results on `CustomGuardBenchmark`.**

Table 2 shows the results when using RoBERTa$_{\text{Large}}$ as the base model, which unlike Multilingual-E5$_{\text{Large}}$-Instruct has not been pretrained specifically for high performing sentence embeddings, nor has it been further pretrained with an instruct-based corpus. Due to this we see a drop in performance, however, we are still within 0.56 average F1 score points compared to 69.41 F1 obtained by `gpt-4` in Table 1. Moreover, all other baselines are outperformed and significant improvements are found when using our synthetic training data compared to the real data training data that is available from each public dataset. Additionally, `MultiTaskGuard` consistently outperforms `TaskGuard` as we posit the effects of GIP in `MultiTaskGuard` has more impact than Multilingual-E5$_{\text{Large}}$-Instruct since it has not been pretrained with instructions prior to GIP.

**Private Benchmark Results** From Table 3, we find that `UniGuard` demonstrates superior performance across all categories of the `CustomGuardBenchmark`[3]. UniGuard consistently outperforms strong baselines, including `gpt-4` and other SoTA models, with an increase 5.48 F1 score points over `gpt-4` (89.52 vs. 84.04 average across all categories). This is a result of using TIES model merging of base model parameters combined with Thompson sampling search. `UniGuard` performance is particularly noteworthy in the Safety and Injection categories, where it achieves the highest scores of 91.83 and 88.62, respectively. While `gpt-4` is competitive in performance for safety and prompt injection, it suffers in performance on more specialized guardrail tasks, namely in `Finance` (i.e prohibiting financial advice) and to a lesser extent `Tax` (i.e Avoid Tax Advice).

**`MultiTaskGuard` requires less task-specific fine-tuning** During our experiments we found that `MultiTaskGuard` classification layer fine-tuning (CFT) *outperforms full fine-tuning* (FFT) while `TaskGuard` *requires FFT* for optimal performance. This can be observed from our results in Figure 3. Across each task of `CustomGuardBenchmark` we find that in fact `TaskGuard` heavily relies on FFT to generalize well, particularly on the "Avoid financial advice" and "Avoid Unsafe Discussions" policies. In contrast, on average, the F1 is higher with CFT compared to FFT for `MultiTaskGuard`. From these results, we conclude that GIP plays a vital role in generalizing well to novel (unseen) policies such as those corresponding to tasks within `CustomGuardBenchmark` and requires only few-shot samples to obtain a slight generalization increase for optimal performance.

**`MultiTaskGuard` needs less fine-tuning data to generalize well** Not only do we find that less active parameters (i.e classification layer only) are required for optimal performance, but also less training samples. Figure 4 shows the F1 scores after fine-tuning `TaskGuard` (no GIP) and `MultiTaskGuard` (with GIP) with an increasing number of training samples across `Safety` and `Finance` `CustomGuardBenchmark` test sets. We find that not only does `MultiTaskGuard` also converges quicker than `TaskGuard` as for these experiments the average number of epochs require to train per task is 1 for `MultiTaskGuard` and

---

[3] `CustomGuardBenchmark` will be made public at https://huggingface.co/datasets

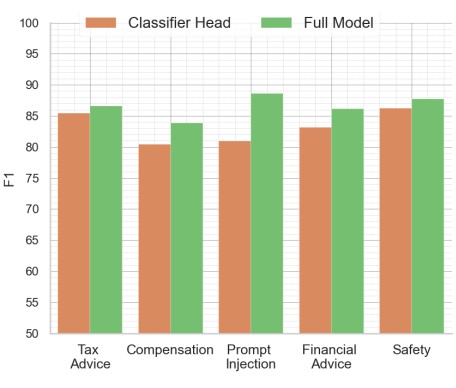

(a) **TaskGuard**

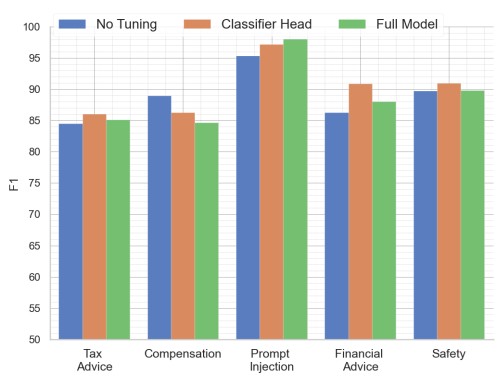

(b) **MultiTaskGuard**

Figure 3: **Model Performance Differences of Classifier-Only vs Full Model Tuning**

8 for `TaskGuard`. Moreover, it is also observed that `MultiTaskGuard` performance is nearly on par without any additional task-specific fine-tuning. Hence, the zero-shot performance and generalization to new unseen guardrailing policies/tasks has been drastically improved due to our GIP step on synthetic guardrail data. Moreover, `MultiTaskGuard` zero-shot performance exceeds the baseline LLMs from Table 1, 2 and 3.

**Model Merging Ablation Results**
Table 4 shows SLERP, TIES, DARE and MSA$_{Average}$ when used with and without our proposed MMS for improve guardrailing. These results show the use of Thompson sampling for Bayesian search of the optimal top-$k$ model weightings. We find that in all cases, the use of MMS to produce UniGuard improves results when the number of search iteration is increase from $T = 1 \rightarrow 50$. We increase 0.55 F1 on SafeGuard (prompt-injection) using TIES, 0.31 F1 on ToxicChat (toxicity) using SLERP and 0.68 F1 on NVIDIA-CS (safety) using TIES. In all cases, increasing the num-

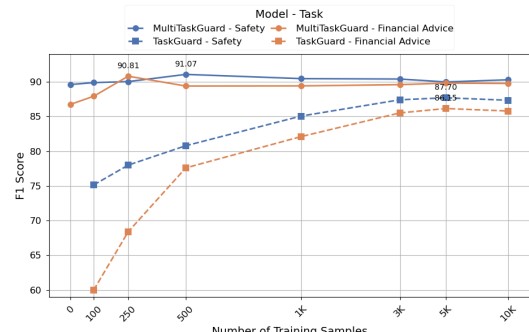

Figure 4: `TaskGuard` & `MultiTaskGuard` **Learning Curves** on `Safety` and `Finance` test sets.

ber of MMS iterations leads to improved generalization. After 50 iteration we find F1 scores plateaued across all benchmarks. Moreover, Thompson sampling consistently improves over random search for the optimal weight combinations for each model merging algorithm. We also find that on average the attention-only parameters or base model parameters are the best choice for MMS and using it with the top-1 models embeddings and classification layer.

## 6 CONCLUSION

This work proposed a process for producing high performing classifiers that generalize well the custom policies that define the scope of a guardrail. We find that with models that are less than 1GB in storage we can outperform models of magnitudes of order larger, such as `gpt-4`, by 21.62 F1 points and outperform well-established and publicly available guardrails, such as those from the LlamaGuard suite, by 29.92 points. This was achieved via our proposed guardrail instruction pretraining and then

| Model | Iter. | SafeGuard | ToxicChat | NVIDIA-CS |
|---|---|---|---|---|
| `TaskGuard` | - | 92.73 | 81.39 | 81.65 |
| TIES | 1 | 96.11 | 96.41 | 85.33 |
| SLERP | 1 | 95.68 | 96.41 | 85.33 |
| DARE | 1 | 95.82 | 95.49 | 84.90 |
| MSA$_{Average}$ | 1 | 95.62 | 96.13 | 85.40 |
| TIES | 50 | **96.66** | 97.18 | **86.01** |
| SLERP | 50 | 96.29 | **96.72** | 85.72 |
| DARE | 50 | 95.89 | 96.20 | 85.75 |
| MSA$_{Average}$ | 50 | 96.48 | 96.82 | 85.94 |

Table 4: **Comparison of Model Merging Techniques for Guardrailing.**

further improved with our model merging search. Our guardrail models require relatively less training data and active fine-tuning parameters to adapt to new policies. We view this as a breakthrough for faster, customizable and low cost guardrailing of general purpose large language models and on-device given the reduced memory and storage footprint.

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
