# OpenReview forum: "Unified Multi-Task Learning & Model Fusion for Efficient Language Model Guardrailing"
_ICLR.cc/2025/Conference — Submitted to ICLR 2025_

### Official Review · Reviewer_TbvE · 2024-11-01

**Soundness:** 2
**Presentation:** 2
**Contribution:** 3
**Rating:** 3
**Confidence:** 3

**Summary:**

The paper introduces a method called TaskGuard for language model guardrailing that uses synthetic data to finetune a small model like RoBERTa and achieves SOTA performances. They use a single model that is pretrained on a large multitask synthetically generated dataset, called MultiTaskGuard, to further improve the generalization. Furthermore, they propose UniGuard, a search-based model merging approach that finds an optimal set of parameters to combine TaskGuard models and MultiTaskGuard models. Experimental results show that when equipped with the proposed method, small models like Roberta even outperform LlamaGuard and GPT-4o on 7 public datasets and 4 internal datasets.

**Strengths:**

1.	The method could achieve SOTA guardrailing performance against large language models like GPT-4o and LlamaGuard.

2.	Based on only a sub 1GB classifier and synthetic data, the method could perform very well without the high demand for resources and computation.

**Weaknesses:**

1.	In section 3.1, the authors introduce the process of synthetic data generation formally. However, the generation details are not provided in this section. For example, what is the meta prompt of LLM used for data generation? What is the template that prompts LLM to self-reflect on its own label judgments? Is there any example that shows the synthetic data? This kind of information is essential for future reproduction.

2.	It is not clear whether MultiTaskGuard and TaskGuard share the same task format and input schema. If not, it is somewhat strange that UniGuard combines its best-performing models tuned towards different input schemas. If so, more ablation studies should be conducted to test the performance where only MultiTaskGuard models are merged or only TaskGuard models are merged.

3.	Table 1 shows the outstanding performance of the proposed methods. From the table and the experimental settings, I can infer that the API guard models and open guard LLM-based guard models are tested in zero-shot. In contrast, TaskGuard and MultiTaskGuard are trained on real data or synthetic data tailored to specific benchmarks. In this way, the comparison setting may be unfair. I am curious about how a large language model, e.g., Llama2-7B, would perform when tuned on the training data.

4.	The extensibility of MultiTaskGuard and UniGuard is unclear. When more policies come in, the trained model may fail to guard new cases. How to incrementally add new policies without influencing the original performance needs to be further discussed.

**Questions:**

See above.

---

> ### Author Response · Authors · 2024-11-23
> **Addressing feedback from Reviewer TbvE**
>
> Thanks you for reviewing the paper, your feedback is valuable.
>
> Weaknesses:
>
> Q1 - We actually have this information in the appendix (it was not attached unfortunately). Since we have ample space in the paper, we are moving the templates used for generation into the methodology section. This also include the prompt template we used for self-reflection and an example output from this process.
>
> Q2 - The input schema for MultiTaskGuard follows from that described in Sec. 3.3 - $\bar{x}_i = \text{Instruct: }\mathcal{P}_{{(i,\text{desc})}} \text{ [SEP]\textbackslash n Query: } x_i \text{ [SEP] } r_i$.
>
> At inference it is $\bar{x}_i = \text{Instruct: } \mathcal{P}_{{(i,\text{desc})}} \text{ [SEP]\textbackslash n Query: } x_i$.
>
>  For TaskGuard fine-tuning and inference it is $\bar{x}_i = \text{Instruct: } \mathcal{P}_{{(i,\text{desc})}} \text{ [SEP]\textbackslash n Query: } x_i$.
> Hence, you can see the inference schema is the same for TaskGuard and MultiTaskGuard which allows us to merge them as long as the instructions for both are the same so the merged models can use the same instruction. We actually have created a module for optimizing the instruction when the instructions are different during MMS. We will describe these details in the methodology section.
>
> Q3 - We actually have the results of the LLMs fine-tuned but the results did not improve much and Aegis-LlamaGuard out of the box with the same policy definition was still the best on average. We can put these fine-tuned LLM performance in the final version of the paper. However, we still believe the baselines should be the original models as the fine-tuning process we are proposing with our synthetic data pipeline is apart of our contribution in this paper.
>
> Q4 - MultiTaskGuard can easily adapted to new incoming policies as GIP has led to a very generalizable base model, as reflected in our results section particularly in Figure 3 and Figure 4.

---

### Official Review · Reviewer_MXFk · 2024-11-01

**Soundness:** 3
**Presentation:** 3
**Contribution:** 3
**Rating:** 8
**Confidence:** 3

**Summary:**

In this paper the authors propose a new method, using multi-armed bandit to merge guardrailing across multiple policies. Essentially, the trained model is able to handle multiple guardrail tasks (policies). It can better generalize to new policies, and requires less fine tuning data and fine tuning parameters.
The benchmarking results over other guardrailing models show superior performance.

The new model, MultiTaskGuard, is a multi class detector/classifier, it is not clear from the paper what information it outputs per class (for example per class confidence level) if at all, or it merges and considers the multi-policies as one new policy.

**Strengths:**

New technically sound approach for merging multiple guardrails across multiple policies into one model.

**Weaknesses:**

Doesnt explain how multi class classification is being handled and what output is being provided and what is the performance per different policy in the fused MultiTaskGuard model.

**Questions:**

It is not clear from the paper if the MultiTaskGuard provides confidence levels for the multi class classification, for each considered policy?

---

> ### Author Response · Authors · 2024-11-23
> **Addressing feedback from Reviewer MXFk**
>
> Thanks you for taking the time to review this paper, much appreciated.
>
> Weaknesses:
> Doesnt explain how multi class classification is being handled and what output is being provided and what is the performance per different policy in the fused MultiTaskGuard model.
> Questions:
> It is not clear from the paper if the MultiTaskGuard provides confidence levels for the multi class classification, for each considered policy?
>
> Response to both Weaknesses and Questions
>
> - The output of MultiTaskGuard is a single binary classification head, the "MultiTask" part here is because of the fact the instruction-prompt pairs are generated from various policy definitions (101k unique policies of 1.2 million samples in the pretraining dataset). Hence, for all policies/tasks we share the same binary classification layer, what distinguishes whether the prompt is safe or not even though we use the same classification layer is the instruction we use which is derived from the policy description, allowed and disallowed behaviors.
> - The performance per task is displayed in Table 1 - 3.

---

### Official Review · Reviewer_WyTq · 2024-11-03

**Soundness:** 2
**Presentation:** 1
**Contribution:** 2
**Rating:** 1
**Confidence:** 3

**Summary:**

The paper proposes a unified framework for guardrailing large language models (LLMs) to ensure safety and efficiency when filtering out unsafe or malicious content. The main contributions include the development of a synthetic data generation pipeline, a multi-task learning approach called MultiTaskGuard, and a model merging search strategy to optimize guardrailing models.

**Strengths:**

- the paper addresses an important problem.

**Weaknesses:**

- the paper is hard to follow: (1) it's rare the say a model is a "1GB" classifier; (2) using unnecessary abbreviations (e..g., synthetic data generation => SDG) can be misleading; (2) inconsistently using unnecessary math symbols can be even more misleading (e.g.,  $P_{\text{name}}$ denotes the name attribute of the polices, but $P_{i}$ refers to the i-th police); and (3) incorrectly using of escape characters (e.g., in line 169 "\n").

- related work is not cited in a proper way. For example, in lines 258-265, the paper provides the links to previous work but does not cite them.

- There is no appendix in the paper, but some details are said to be provided in the appendix.

- the proposed methods, including synthetic data generation and model merging, are not introduced clearly.

- it is not clear how are the baseline models such as gpt-4 adopted for the experiments. For example, how does the prompt look like? Are few-shot examples included? Moreover, there is no specific version of the GPT models, making it impossible to make reliable comparisons between different models.

- none of the tables are captioned in a reasonable way

**Questions:**

- where is the appendix?
- will you public the new test set?

---

> ### Author Response · Authors · 2024-11-23
> **Addressing feedback from Reviewer WyTq**
>
> Thanks for your feedback.
>
> Weaknesses:
> - Thanks for pointing out the inconsistency in subscript notation we will fix this. As for the SDG abbreviation, we can change if you think it improves readability, we just though the repeating synthetic data generation multiple times throughout the paper was a slight bit cumbersome.
>
> - We have now put the citations for the public dataset and have included the appendix.
> - We use the sample policy definition as a prompt template for gpt-4 and other LLM models. We have now put a diagram in the paper and a short paragraph explaining how this used at inference time. Also in all the text and tables we have included the full version extension to each gpt baseline.
> - none of the tables are captioned in a reasonable way - we can move the captions above the Tables instead of below if this is what you're referring to.
>
> Questions:
> will you public the new test set? - Yes our new benchmark test sets will be made public with a full readme describing dataset statistics, policy definitions used to generate the data and generation configurations used in our synthetic data generation pipeline.

---

### Official Review · Reviewer_qs3W · 2024-11-04

**Soundness:** 2
**Presentation:** 2
**Contribution:** 2
**Rating:** 3
**Confidence:** 3

**Summary:**

The paper introduces three models: TaskGuard, MultiTaskGuard, and UniGuard. TaskGuard is trained on synthetic data tailored to specific tasks, MultiTaskGuard is trained on multi-task synthetic data for broader applicability, and UniGuard combines TaskGuard and MultiTaskGuard through model merging techniques. These models achieve state-of-the-art performance across 11 datasets, offering enhanced accuracy and efficiency.

**Strengths:**

1. The proposed models achieve superior accuracy and efficiency compared to existing approaches.

2. The evaluation, conducted on 11 datasets, provides a comprehensive assessment.

**Weaknesses:**

1. Key hyperparameters, such as lambda, k, and n, are not specified.

2. An ablation study examining the impact of different components in the multi-task training loss (line 173) is missing.

3. Data statistics are unclear, specifically for the training, validation, and test sets.

4. The proposed MMS method results in only a modest performance improvement.

**Questions:**

Q1. How many policies are used to train TaskGuard? Is a separate version of TaskGuard, MultiTaskGuard, and UniGuard trained for each dataset? Could you provide examples of the policies used?

Q2. How many training stages are there for MultiTaskGuard, TaskGuard, and UniGuard?

Q3. During inference, what is the input format for TaskGuard, MultiTaskGuard, and UniGuard?

Q4. What are the ablation results for multi-task training loss?

Q5. What is the over-refusal rate of the proposed model on xstest?

Q6. How does the model merging cost of the proposed method compare to the baseline?

Q7. For the public benchmark, was training data utilized in the data synthesis process?

Q8. What backbone model is used in Table 3?

---

> ### Author Response · Authors · 2024-11-23
> **Addressing feedback from Reviewer qs3W**
>
> First of all, thank you very much for your feedback. Below addresses the point raised.
>
> Weaknesses:
>
> 1. We have now specified these hyperparameters in the relevant section.
> 2. Fair point, as in the response below to Q4, we have results to provide on this and will put in a final version.
> 3. We will elaborate on training, val and test sets used e.g number of safe/unsafe samples, policy definition used for their generation, configurations that were set that dictate if they are borderline, in-domain, slightly out of domain and whether syntactic augmentation were used to test guardrailing robustness to noise.
> 4. It is true that the gains made from MMS are less so than gain made from going from TaskGuard -> MultiTaskGuard using our guardrail instruction pretraining. However, the cost of MMS is less w.r.t training as it only requires an evaluation on a validation set to search for optimal weights in model merging step.
>
> Questions:
>
> Q1 - TaskGuard is trained on the a single policy (the same same policy we test on the benchmark results). MultiTaskGuard is trained on 101,000 unique policies generated with a large LLM and UniGuard potentially combines TaskGuard (policy-specific) and MultiTaskGuard models together but is not technically trained on anything (i.e no gradient updates in the model merging step).
>
> Q2 - There is 1 training stage for MultiTaskGuard and TaskGuard and no training of UniGuard as described on Q1.
>
> Q3 - The input format at inference is "Instruct: <policy based instruction>\nQuery: <prompt>"
>
> Q4 - We do have results of when training is done only with MLM, MLM + classification loss and MLM + classification loss + AlicePP loss if thats what you mean. We can put these ablation results in the paper if accepted.
>
> Q5 - The average refusal rate across prompt types in xtest is 5.8% for safe prompts and for unsafe prompts 98% for unsafe prompts. We can also include these results in a final version if accepted.
>
> Q6 - Our model merging search is extremely quick and can be parallelized and can be used with the baseline model merging techniques.
>
> Q7 - Yes, all results in our Tables that have $_{\text{Synthetic}}$ correspond to models that were trained on synthetic data using a policy definition that describes the task, as described in our methodology section.
>
> Q8 -  The backbone is an instruction-tuned version of XLM-RoBERTa-Large, described in our experimental setup.

---

### Meta-Review · Area_Chair_CW9r · 2024-12-20

**Metareview:**

This paper introduces three models, TaskGuard, MultiTaskGuard, and UniGuard, for guardrailing large language models (LLMs) to ensure safety and efficiency. TaskGuard focuses on single-policy synthetic data, MultiTaskGuard extends this with multi-policy data, and UniGuard combines the two via model merging techniques.

Strengths
- The paper addresses an important and timely challenge in AI safety—guardrailing LLMs against unsafe and malicious content.

Weaknesses:
- Many critical details are missing: for example,  on synthetic data generation, prompts, and templates are inadequately explained, and reproducibility relies on rebuttal clarifications.
-Claims of state-of-the-art performance are based on unclear and possibly unfair comparisons.

**Additional Comments On Reviewer Discussion:**

While the authors claim state-of-the-art performance on 11 datasets, concerns about the methodology, presentation, and experimental fairness remain significant. Multiple reviewers highlighted issues with clarity, missing implementation details, and unfair experimental comparisons, alongside concerns about the methods' extensibility and practical significance. While the authors addressed some points during the rebuttal, the paper's core weaknesses—unclear methodology, biased comparisons, and modest technical contributions—remain unresolved.

---

### Decision · Program_Chairs · 2025-01-22

Reject